# The relationship between ambivalence over the expression of emotions and somatic symptoms among Iranian long-distance and geographically close partners: The mediating role of emotional suppression

**Nazanin Okati**, **Leyla Rangamiztoosi**, **Maryam Gholipour**, **Fariba Zarani***

Department of Psychology, Shahid Beheshti University, Tehran, Iran

* f_zarani@sbu.ac.ir

## Abstract

Previous research has shown that difficulties in emotional expression may be linked to physical health symptoms, but few studies have explored this in the context of romantic relationship types. This study investigates the role of emotional suppression as a mediator between emotional ambivalence and somatic symptoms in long-distance (LDR) and geographically close relationships (GCR). A cross-sectional design was used with a convenience sample of 442 adults currently in romantic relationships, including 215 in LDRs (M_age = 26.8 years; 179 females, 48 males) and 227 in GCRs (M_age = 31.3 years; 187 females, 28 males). Participants completed the Ambivalence Over the Expression of Emotion Questionnaire (AEQ), Emotion Regulation Questionnaire (ERQ), Patient Health Questionnaire-15 (PHQ-15), and the Long-Distance Romantic Relationship Index. Data were analyzed using Structural Equation Modeling (SEM) and multigroup analysis via SmartPLS. There was a significant correlation between negative emotional ambivalence and somatic symptoms in both relationship groups. However, the mediating role of emotional suppression in this relationship was not supported. Gender differences in emotional suppression showed contrasting patterns between LDRs and GCRs. Moreover, participants in LDRs reported significantly higher somatic symptoms compared to those in GCRs. These findings suggest that the link between emotional ambivalence and somatic symptoms may involve other contributing factors beyond emotional suppression. This research highlights the importance of considering relationship context and gender in understanding how emotional experiences affect physical health.

which permits unrestricted use, distribution, and reproduction in any medium, provided the original author and source are credited.

**Data availability statement:** Yes - all data are fully available without restriction; All relevant data are within the paper and its Supporting Information files.

**Funding:** The authors received no specific funding for this work.

**Competing interests:** The authors have declared that no competing interests exist.

## Introduction

Intimate romantic relationships are fundamental human needs that contribute significantly to physical and mental health [1, 2]. Most research on close and intimate relationships has focused on geographically close partners (GCRs), based on the assumption that proximity and face-to-face interactions are essential for emotional attachment and intimacy [3]. However, in recent years, social changes such as globalization, increased mobility, immigration, and virtual communication have made long-distance romantic relationships (LDRs) more common [4, 5]. For example, approximately 3.5 million Americans live apart from their spouses for reasons other than separation or divorce [6], and similar trends have been observed in other countries like the UK, Germany, Hong Kong, and Israel [7].

Despite their physical distance, individuals in LDRs often aim to maintain closeness and commitment. However, they may face unique challenges including limited face-to-face contact [8], changes in roles and responsibilities [9], and reduced sexual intimacy and social support [10]. In contrast, GCRs benefit from frequent in-person interactions that facilitate emotional regulation and support. As a result, individuals in LDRs may be more vulnerable to psychological stress, which in turn may contribute to somatic symptoms [11].

Somatic symptoms—such as pain, dyspnea, heart palpitations, numbness, and digestive issues—are physical complaints often triggered by intense emotions and thoughts, with many symptoms remaining medically unexplained [12, 13]. Difficulties in emotional intimacy and coping with relational loss may exacerbate these symptoms, particularly in LDRs [11]. For instance, Pereira et al. [14] found that 83% of women in LDRs experienced at least one moderate or severe somatic symptom. However, few studies have directly compared the prevalence of somatic symptoms between LDRs and GCRs.

A key factor influencing somatic symptoms is emotional ambivalence [15]—the simultaneous experience of positive and negative emotions toward a partner or relationship [16, 17]. Ambivalence often arises from inevitable conflicts, criticism, personality differences, and disagreements within relationships [18, 19]. Importantly, emotional ambivalence is linked to physiological stress responses such as increased blood pressure and cardiovascular reactivity [20, 21], which may contribute to somatic symptoms [22, 23].

In LDRs, emotional ambivalence may be heightened by the "separation-reunion cycle" [24], where partners alternate between periods of proximity and separation. This cycle can increase positive emotions temporarily while suppressing conflicts and negative feelings, as partners avoid addressing issues to preserve limited face-to-face time [25, 26]. This avoidance often manifests as emotional suppression—deliberate inhibition of emotional expression—which, although an emotion regulation strategy, can negatively impact intimacy, psychological well-being, and increase somatic symptoms [27–29].

Emotional suppression may worsen physical health by fostering unhealthy coping behaviors [30] and activating physiological stress systems, including the

hypothalamic–pituitary–adrenocortical axis and immune disruption, leading to symptoms like heart palpitations and increased susceptibility to infections [31, 32]. Given the established link between negative emotions and somatic symptoms [33], emotional suppression could mediate the relationship between emotional ambivalence and somatic complaints.

While positive relationships generally predict better mental health [34], research has largely overlooked LDRs. Given the potential overlap between emotional ambivalence, suppression, and somatic symptoms, understanding their interplay within the context of relationship type is important. Prior LDR research has often relied on homogeneous samples, mostly White, middle-class couples in Western, individualistic societies [11,24]. This narrow focus may create the misconception that LDRs only exist within these groups, overlooking cultural norms that shape attachment and relational expectations. Examining LDRs in a collectivist context such as Iran is therefore essential, as cultural differences may produce distinct patterns and outcomes compared to Western samples. This study seeks to fill this gap by investigating emotional ambivalence, emotional suppression, and somatic symptoms across these relationship types. Therefore, the present study aimed to investigate: [1] whether somatic symptoms are more prevalent in individuals in LDRs compared to GCRs, [2] whether emotional ambivalence is associated with somatic symptoms in both relationship types, [3] whether emotional suppression mediates this association, and [4] whether the mediation differs by relationship type.

## Method

### Ethics statement

The study was approved by the Ethics Committee of Shahid Beheshti University (Approval No. IR.SBU.REC.1402.078). Inclusion and exclusion criteria were checked before inviting eligible individuals to participate. Participants provided online informed consent and were informed of their right to withdraw at any point without any consequences.

### Study design and sample size estimation

The present study employed a structural equation modeling (SEM) approach to examine the hypothesized relationships. To estimate the required sample size, Cohen's [35] guidelines were used. Considering a maximum of four predictors pointing to a single latent variable in the model, a power level of 0.80, an alpha of 0.01, and a minimum $R^2$ of 0.10, the required sample size for each group (LDRs and GCRs) was calculated to be 191.

### Participants and procedure

A total of 442 individuals currently in romantic relationships participated in this study, including 215 in long-distance relationships (LDRs; 179 females, 48 males; $M$ age = 26.8 years) and 227 in geographically close relationships (GCRs; 187 females, 28 males; $M$ age = 31.3 years). As all items were mandatory in the online survey platform, no missing data were observed. Participants were recruited via convenience sampling through social media platforms (Instagram, WhatsApp, and Telegram) and were provided informed consent electronically prior to completing the online questionnaires. Data were collected using an online questionnaire from June 2023 to January 2024.

The inclusion criteria were: (1) age between 18 and 45 years, (2) being in a committed romantic relationship for at least six months, and (3) informed consent to participate in the study. Participants older than 45 years of age were excluded to minimize the potential influence of age-related health conditions on somatic symptoms. Moreover, younger and middle-aged adults in Iran generally have greater access to social media and online technologies, which was essential for recruitment through online platforms and completing the online questionnaire. In addition, given the recent migration trends in Iran, individuals within this age group were more likely to be involved in long-distance relationships compared to older adults. Therefore, this age range was considered more representative of the population under study. Participants were excluded if they reported a history of diagnosed psychiatric or medical conditions (based on self-report).

To ensure data quality, all survey items were set as mandatory to prevent missing data, and the dataset was reviewed to detect and exclude inconsistent or patterned responses. The authors had access only to participants' email addresses, which were collected solely for the purpose of informing them about the overall study results after publication. Providing email addresses was entirely optional, and participants were informed that sharing their email was voluntary. No other identifiable information was accessed or recorded. A total of 442 participants completed the questionnaire and were classified into long-distance romantic relationships (LDRs) or geographically close romantic relationships (GCRs) based on their scores on the Long-Distance Romantic Relationship Index [36]. As the LDR Index does not provide a standardized cut-off point, participants were categorized using the sample mean, with scores above the mean classified as LDRs and those below or equal to the mean as GCRs.

## Measurements

The study used four self-report questionnaires to assess emotional ambivalence, emotion suppression, somatic symptoms, and relationship type classification.

### The ambivalence over the expression of emotion questionnaire (AEQ)

The AEQ [15] was developed to investigate the importance of ambivalence in emotional expression in health. This questionnaire has 27 items and consists of two subscales: ambivalence in expressing positive emotions such as love and affection (Questions 1–13) and ambivalence over the expression of emotions of entitlement, which can be described as emotions such as anger, pride, and jealousy (Questions 14–27). This instrument is scored 5-point Likert scale ranging from 1 (never) to 5 (always). The scores (no reversal scoring) range from 28 to 140, with higher scores indicating increased ambivalence over emotional expression. Internal consistency for the original scale and subscales ranged from.77 to.89 [15]. The Persian version, validated by Alavi et al. [37], demonstrated excellent reliability (α = .89).

### Emotion regulation questionnaire (ERQ)

The ERQ [38] was designed to assess individual differences in the habitual use of two emotion regulation strategies: cognitive reappraisal and expressive suppression. Answers based on the Likert scale ranged from 1 (strongly disagree) to 7 (strongly agree). The scores in this questionnaire range from 10 to 70. High scores on the expressive suppression subscale indicate a greater tendency to inhibit or suppress the expression of emotions. Considering the research objectives of this study, only the emotional suppression items (questions 2, 4, 6, and 9) were used. Cronbach's alpha coefficient for cognitive reappraisal and expressive suppression was reported as 0.79 and 0.73, respectively, and it also demonstrated a good 2-month test-retest reliability of about 0.7 [38]. This questionnaire was validated in Iran by Qasimpour et al. [39], who reported internal consistency with Cronbach's alpha coefficients ranging from 0.61 to 0.81, demonstrating acceptable reliability of the scale in the Iranian population.

### Patient health questionnaire (PHQ-15)

The PHQ-15 [40] assesses the severity of somatic symptoms using 15 items rated on a 3-point scale: 0 (not bothered at all), 1 (bothered a little), and 2 (bothered a lot). Scores range from 0 to 30, with higher scores reflecting more severe somatic symptom burden. Lee et al. [41] reported good internal consistency for this inventory (α = 0.79). Its Persian version was validated by Abdolmohammadi et al. [42], with a reported alpha of 0.76 and a strong correlation (r = 0.74) with the somatization subscale of the SCL-90, confirming convergent validity.

### Long-distance romantic relationship index

Developed by Pistole and Roberts [36], this 12-item scale assesses relationship type based on responses to a 7-point Likert scale from 1 (strongly agree) to 7 (strongly disagree). Higher scores indicate a long-distance relationship, and lower

scores show a geographically close one. Two additional subscales in this questionnaire were used to assess the validity of the major scale: time together (2 items) and relational importance (3 items). Pistole and Roberts [36] reported satisfactory structural and construct validity and high internal consistency. In a Persian validation study by Bazani et al. [43], the scale showed excellent internal consistency ($\alpha = 0.96$), while the time together and relational importance subscales had alphas of 0.65 and 0.97, respectively.

## Data analysis

Data were analyzed using Partial Least Squares Structural Equation Modeling (PLS-SEM) in SmartPLS 3.0, along with descriptive statistics performed in SPSS 25. PLS-SEM was selected due to its suitability for complex models with small-to-moderate sample sizes and non-normal data distributions [44]. The analysis was carried out in the following steps:

First, descriptive statistics and demographic characteristics were analyzed using SPSS. Next, the measurement model was evaluated to assess reliability and validity. Internal consistency was examined using Cronbach's alpha, Composite Reliability (CR), and the rho_A index, with acceptable thresholds set at ≥ 0.70. Convergent validity was assessed through item loadings (≥ 0.40), Average Variance Extracted (AVE ≥ 0.50), and CR. Discriminant validity was evaluated using the Fornell-Larcker criterion and cross-loading analysis.

Then, the structural model was assessed using the bootstrap resampling method with 500 subsamples to determine the significance of path coefficients. Model quality was further evaluated using coefficient of determination ($R^2$).

Finally, a Multi-Group Analysis (MGA) was conducted to compare structural paths between participants in long-distance relationships (LDRs) and geographically close relationships (GCRs). Group classification was based on scores from the Long-Distance Romantic Relationship Index [36], where higher scores indicated LDRs and lower scores indicated GCRs. All raw data underlying these analyses are provided in S1 Data (CSV) and S2 Data (SPSS), with detailed descriptions of the variables available in S1 Text.

## Results

### Sample characteristics and descriptive statistics

Table 1 presents the demographic and descriptive statistics for the participants.

The results of Table 1 show that the average age of couples in a long-distance relationship (26.86 years) was significantly lower ($t = 7.876$, $p < 0.001$) compared to GCRs (31.35 years). The gender distribution in the two groups was significantly different ($\chi^2 = 5.116$, $p = 0.024$), showing that over two-thirds of the participants were women. Regarding relationship duration, 42.9% of participants in both groups reported being in a relationship for 1–3 years. Couples in LDRs reported significantly lower scores on the "time spent together" item of the Long-Distance Romantic Relationship Index ($M = 8.67$) than GCRs ($M = 11.14$), ($t = 8.581$, $p < 0.001$). Relationship importance scores were also lower in LDRs compared to GCRs ($t = 2.195$, $p = 0.029$).

In terms of the variables, there was a significant difference between the somatic symptoms in the two groups ($t = 9.323$, $p = 0.025$). Although the average of positive and negative emotional ambivalence in LDR couples was higher than in GCRs, this difference was not statistically significant ($p > 0.05$). There was no significant difference in emotional suppression between the two groups of couples ($t = 9.323$, $p = 0.025$).

### Measurement model

The internal consistency of the factors was assessed using Cronbach's alpha coefficient, composite reliability, and rho_A index, as presented in Table 2.

**Table 1. Participants' demographic and descriptive statistics.**

| Variables | Distance | | Total (1247) | statistic (P-Value) |
|---|---|---|---|---|
| | Close (227) | Long (215) | | |
| Age (Years) | 31.349 | 26.859 | 29.043 | 7.876†***(<0.001) |
| Gender | | | | 5.116‡**(0.024) |
| Female | 187(87%) | 179(78.9%) | 366(82.8%) | |
| Male | 28(13%) | 48(21.1%) | 76(17.2%) | |
| Duration of Relationship (Years) | | | | 54.44‡***(<0.001) |
| <1 year | 13(6.2%) | 45(21.3%) | 58(13.8%) | |
| 1-3 years | 75(35.9%) | 105(49.8%) | 180(42.9%) | |
| 4-6 years | 41(19.6%) | 38(18.0%) | 79(18.8%) | |
| 7-10 years | 48(23.0%) | 15(7.1%) | 63(15.0%) | |
| >10 years | 32(15.3%) | 8(3.8%) | 40(9.5%) | |
| Long Distance | 12.291 | 44.451 | 27.934 | 54.671†***(<0.001) |
| Time Together | 11.14 | 8.674 | 9.873 | 8.581†***(<0.001) |
| Relationship Importance | 24.149 | 23.154 | 23.638 | 2.195†**(0.029) |
| Somatic Symptoms | | | | 9.323‡**(0.025) |
| None-minimal | 37(17.2) | 30(13.2) | 67(15.2) | |
| Mild | 82(38.1) | 64(28.2) | 146(33) | |
| Moderate | 56(26) | 70(30.8) | 126(28.5) | |
| Severe | 40(18.6) | 63(27.8) | 103(23.3) | |
| Emotional Ambivalence | | | | |
| Positive Ambivalence | 33.0 | 34.78 | 33.919 | -1.772† (0.077) |
| Negative Ambivalence | 42.51 | 44.22 | 43.391 | -1.927† (0.055) |
| Emotional Suppression | 12.81 | 12.95 | 12.889 | -0.245† (0.807) |

Notes: ‡ statistic, † t statistic, **significance at 0.05, ***significance at 0.01.

**Table 2. Quality criteria.**

| Group | Construct | Cronbach's Alpha | rho_A | CR | AVE |
|---|---|---|---|---|---|
| GCRs | | | | | |
| | Emotional Suppression | 0.821 | 0.829 | 0.882 | 0.653 |
| | Negative Emotional Ambivalence | 0.828 | 0.839 | 0.859 | 0.308 |
| | Positive Emotional Ambivalence | 0.892 | 0.894 | 0.910 | 0.438 |
| | Somatic Symptoms | 0.836 | 0.847 | 0.867 | 0.308 |
| LDRs | | | | | |
| | Emotional Suppression | 0.830 | 0.833 | 0.887 | 0.663 |
| | Negative Emotional Ambivalence | 0.821 | 0.828 | 0.853 | 0.297 |
| | Positive Emotional Ambivalence | 0.885 | 0.886 | 0.904 | 0.423 |
| | Somatic Symptoms | 0.832 | 0.854 | 0.864 | 0.306 |

As shown in Table 2, all constructs demonstrated acceptable internal consistency (Cronbach's alpha > .80). However, AVE values for some constructs, such as positive and negative emotional ambivalence and somatic symptoms, were below the acceptable threshold (.50). To address this, a modified measurement model was tested. Inappropriate items were removed, and the model was re-evaluated. The outcomes of the modified measurement model are presented in Table 3.

 

**Table 3. Modified Measurement Model.**

| Indicators | GCRs | | | LDRs | | |
|---|---|---|---|---|---|---|
| | Factor Loading | T | P-Value | Factor Loading | t | P-Value |
| A1 | 0.614 | 11.689 | <0.001 | 0.655 | 11.497 | <0.001 |
| A2 | 0.735 | 16.976 | <0.001 | 0.737 | 19.381 | <0.001 |
| A3 | 0.801 | 32.423 | <0.001 | 0.721 | 17.196 | <0.001 |
| A4 | 0.733 | 18.353 | <0.001 | 0.762 | 19.512 | <0.001 |
| A5 | 0.779 | 23.829 | <0.001 | 0.739 | 18.446 | <0.001 |
| A6 | 0.688 | 17.512 | <0.001 | 0.633 | 9.325 | <0.001 |
| A7 | 0.660 | 13.567 | <0.001 | 0.600 | 11.981 | <0.001 |
| A8 | 0.749 | 22.428 | <0.001 | 0.709 | 15.753 | <0.001 |
| A9 | 0.672 | 14.948 | <0.001 | 0.690 | 17.751 | <0.001 |
| A18 | 0.721 | 19.127 | <0.001 | 0.721 | 19.088 | <0.001 |
| A24 | 0.750 | 20.398 | <0.001 | 0.678 | 13.804 | <0.001 |
| A25 | 0.721 | 15.782 | <0.001 | 0.632 | 10.329 | <0.001 |
| A26 | 0.725 | 16.494 | <0.001 | 0.582 | 9.256 | <0.001 |
| A27 | 0.720 | 18.298 | <0.001 | 0.753 | 22.298 | <0.001 |
| E1 | 0.824 | 28.326 | <0.001 | 0.834 | 38.585 | <0.001 |
| E2 | 0.785 | 20.021 | <0.001 | 0.781 | 25.701 | <0.001 |
| E3 | 0.884 | 48.398 | <0.001 | 0.847 | 26.679 | <0.001 |
| E4 | 0.700 | 13.195 | <0.001 | 0.791 | 26.616 | <0.001 |
| S6 | 0.723 | 9.556 | <0.001 | 0.705 | 11.543 | <0.001 |
| S7 | 0.675 | 10.625 | <0.001 | 0.673 | 10.988 | <0.001 |
| S8 | 0.747 | 16.077 | <0.001 | 0.672 | 9.699 | <0.001 |
| S9 | 0.772 | 15.604 | <0.001 | 0.796 | 17.242 | <0.001 |
| S10 | 0.763 | 14.615 | <0.001 | 0.735 | 16.430 | <0.001 |

Table 3 showed the factor loadings for the modified measurement model. All retained items had loadings above .40 and were statistically significant ($p < .001$). Table 4 confirms that after model refinement, AVE values improved, and all constructs demonstrated acceptable reliability and convergent validity.

## Structural model

Table 5 demonstrates the results of the structural model evaluation.

Bootstrap analysis using 500 samples shows that in both groups of partners, negative emotional ambivalence had a significant correlation with emotional suppression (GCRs/ $\beta = 0.397$, $p < 0.001$; LDRs/ $\beta = 0.334$, $p < 0.001$) and somatic symptoms (GCRs/$\beta = 0.152$, $p = 0.072$; LDRs/$\beta = 0.227$, $p = 0.005$). Positive emotional ambivalence was also significantly associated with emotional suppression in both groups (GCRs/B = 0.4, $p < 0.001$; LDRs/ $\beta = 0.369$, $t = 5.376$, $p < 0.001$). Emotional suppression did not significantly mediate the relationship between emotional ambivalence and somatic symptoms in either group ($p > .05$). To analyze the coefficient determination, emotional suppression, and positive and negative emotional ambivalence explained the changes in somatic symptoms in GCRs and LDRs as 16.5% and 14.5% respectively. It also showed that changes in emotional suppression in GCRs and LDRs are expressed through positive and negative emotional ambivalence as 39.4% and 51.9% respectively (see Fig 1 for the structural model of geographically close partners and Fig 2 for the structural model of long-distance partners).

Table 6 displays the variance in path coefficient estimation for two comparisons (LDRs and GCRs) and provides the outcomes of multigroup comparisons utilizing the parametric approach.

**Table 4. Reliability & convergent validity of constructs.**

| Latent Variables | Criteria | GCRs | LDRs |
|---|---|---|---|
| Emotional Suppression | Cronbach's Alpha | 0.812 | 0.830 |
| | rho_A | 0.827 | 0.832 |
| | CR | 0.877 | 0.887 |
| | AVE | 0.642 | 0.662 |
| Negative Emotional Ambivalence | Cronbach's Alpha | 0.779 | 0.703 |
| | rho_A | 0.781 | 0.716 |
| | CR | 0.849 | 0.807 |
| | AVE | 0.529 | 0.457 |
| Positive Emotional Ambivalence | Cronbach's Alpha | 0.881 | 0.866 |
| | rho_A | 0.888 | 0.867 |
| | CR | 0.904 | 0.894 |
| | AVE | 0.514 | 0.485 |
| Somatic Symptoms | Cronbach's Alpha | 0.792 | 0.765 |
| | rho_A | 0.802 | 0.777 |
| | CR | 0.856 | 0.841 |
| | AVE | 0.543 | 0.515 |

**Table 5. Results of the structural model evaluation.**

| Path Relationship | | Close-Distance | Long-Distance |
|---|---|---|---|
| *Direct Effect* | | | |
| Emotional Suppression -> Somatic Symptoms | | 0.151 | 0.019 |
| Negative Emotional Ambivalence -> Emotional Suppression | | 0.397** | 0.334** |
| Negative Emotional Ambivalence -> Somatic Symptoms | | 0.152* | 0.227** |
| Positive Emotional Ambivalence -> Emotional Suppression | | 0.400** | 0.369** |
| Positive Emotional Ambivalence -> Somatic Symptoms | | 0.096 | 0.137 |
| Gender -> Emotional Suppression | | -0.036 | 0.099 |
| Gender -> Negative Emotional Ambivalence | | -0.021 | -0.002 |
| Gender -> Positive Emotional Ambivalence | | -0.078 | 0.039 |
| Gender -> Somatic Symptoms | | -0.188** | -0.189** |
| *Indirect Effect* | | | |
| Negative Emotional Ambivalence -> Emotional Suppression -> Somatic Symptoms | | 0.060 | 0.006 |
| Positive Emotional Ambivalence -> Emotional Suppression -> Somatic Symptoms | | 0.061 | 0.007 |
| $R^2$ | Emotional Suppression | 0.519 | 0.394 |
| | Somatic Symptoms | 0.165 | 0.145 |

Notes: *Significance at 0.10, **significance at 0.05, ***significance at 0.01.

Based on the comparison, a statistically significant difference was found only in the path from gender to emotional suppression, indicating a group-level distinction. In GCRs, suppression was more prevalent among men, while in LDRs it was more prevalent among women. Although not significant within each group independently, the difference in path coefficients was statistically meaningful across the groups.

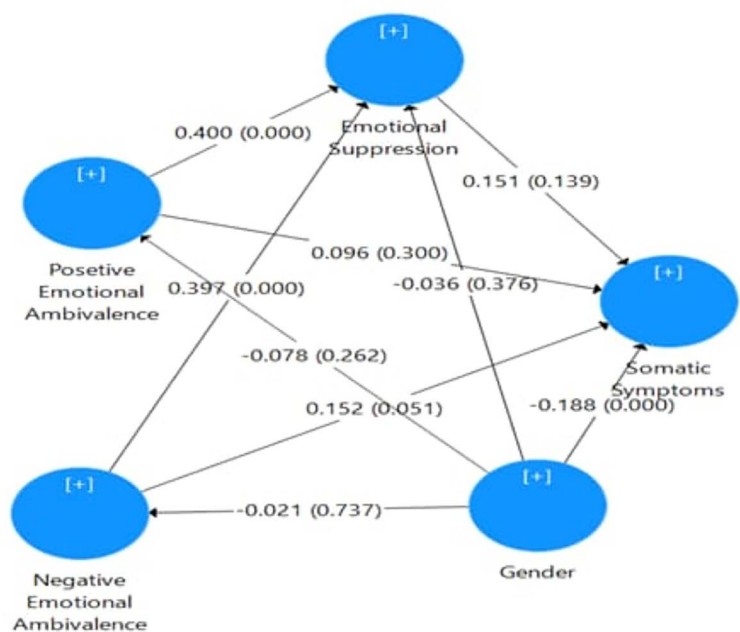

**Fig 1. Diagram of the structural model of geographically close partners.**

## Discussion

The present study investigated whether emotional suppression mediates the relationship between emotional ambivalence and somatic symptoms in both LDRs and GCRs. The results showed that although emotional suppression was significantly predicted by emotional ambivalence in both groups, it did not mediate the relationship between ambivalence and somatic symptoms. However, emotional ambivalence—particularly negative emotional ambivalence—was significantly associated with somatic symptoms in both relationship types. Additionally, the study provided further insights into the role of gender in emotional suppression, highlighting how this association may vary depending on the type of romantic relationship.

As expected, ambivalence over expressing negative emotions was significantly associated with somatic symptoms in both LDRs and GCRs. This finding aligns with previous research suggesting that emotion regulation difficulties can either mitigate or exacerbate physical health outcomes [45]. For instance, Maroti et al. [46] found that emotional awareness and expression therapy significantly reduced somatic symptoms, while Schnabel et al. [23] emphasized the role of emotional clarity and self-efficacy in managing such symptoms. In contrast, no significant association was found between ambivalence over positive emotions and somatic symptoms in either relationship group. One possible explanation is that the habitual suppression of positive emotions may be more culturally normative and less psychologically costly in the Iranian population, thereby weakening its association with somatic symptoms [47, 48]. Moreover, relationship type (LDR vs. GCR) did not moderate the association between emotional ambivalence and somatic symptoms, which supports findings from previous studies showing comparable health outcomes across these relational contexts [49, 50].

Contrary to expectations, emotional suppression did not mediate the relationship between emotional ambivalence and somatic symptoms in either group. Although suppression is often associated with negative cognitive patterns and reduced emotional engagement [12], its effect may have been overshadowed by the broader impact of emotional ambivalence. Ambivalence over emotional expression is a more complex and trait-like construct, often linked to

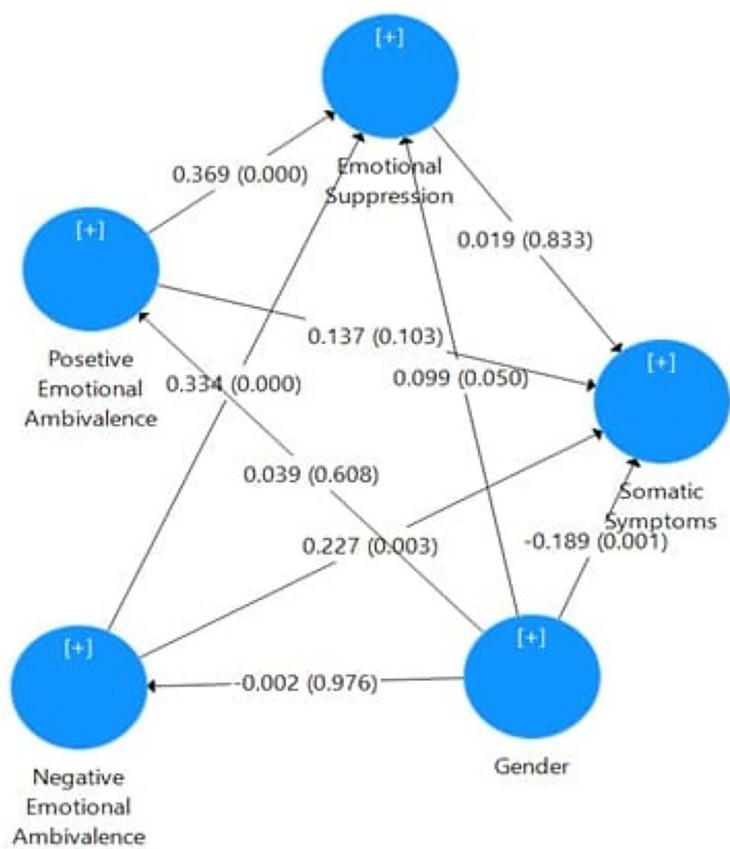

**Fig 2. Diagram of the structural model of long-distance partners.**

**Table 6. Multigroup comparison test results.**

| Relationship | Close-Distance vs. Long-Distance | | | |
|---|---|---|---|---|
| | \|diff\| | t | P-Value | Significance |
| Emotional Suppression -> Somatic Symptoms | 0.132 | 0.957 | 0.339 | Nsig. |
| Negative Emotional Ambivalence -> Emotional Suppression | 0.062 | 0.681 | 0.496 | Nsig. |
| Negative Emotional Ambivalence -> Somatic Symptoms | 0.075 | 0.644 | 0.520 | Nsig. |
| Positive Emotional Ambivalence -> Emotional Suppression | 0.031 | 0.308 | 0.758 | Nsig. |
| Positive Emotional Ambivalence -> Somatic Symptoms | 0.042 | 0.326 | 0.744 | Nsig. |
| Gender -> Emotional Suppression | 0.135 | 2.066 | 0.039** | Sig. |
| Gender -> Negative Emotional Ambivalence | 0.018 | 0.192 | 0.848 | Nsig. |
| Gender -> Positive Emotional Ambivalence | 0.117 | 1.131 | 0.258 | Nsig. |
| Gender -> Somatic Symptoms | 0.001 | 0.016 | 0.987 | Nsig. |
| *Indirect Effect* | | | | |
| Negative Emotional Ambivalence -> Emotional Suppression -> Somatic Symptoms | 0.054 | 1.026 | 0.305 | Nsig. |
| Positive Emotional Ambivalence -> Emotional Suppression -> Somatic Symptoms | 0.054 | 0.953 | 0.341 | Nsig. |

Notes: **significance at 0.05.

personality dimensions such as neuroticism [51], whereas emotional suppression is typically considered a state-based strategy. Research has shown that ambivalence tends to correlate more strongly with psychosocial and health-related outcomes compared to suppression [52, 53]. A key feature of ambivalence is the internal psychological conflict it creates. In the context of long-distance relationships, individuals may struggle more with decisions about whether and how to express emotions due to factors such as physical separation and limited communication opportunities. This conflict may affect their psychological well-being more profoundly than the use of suppression alone [54]. Given its deeper personality roots, ambivalence may be a more influential predictor of somatic symptoms, and future studies should explore this dynamic further.

Gender-related patterns in emotional suppression have traditionally shown that men tend to suppress emotions more than women, often due to societal expectations tied to masculinity [55]. However, our findings challenge this trend. While men in geographically close relationships exhibited slightly more suppression, this pattern was reversed in long-distance relationships—where women reported higher levels of emotional suppression than men. This suggests that emotional suppression is not inherently more prevalent in men but rather influenced by contextual factors such as relationship type, social roles, and personality traits [56].

Although no previous study, to our knowledge, has explored this gendered difference specifically within LDRs versus GCRs, several factors may help explain it. In LDRs, women may adopt more self-reliant emotional strategies due to physical separation, often managing challenges independently in their partner's absence [57]. The lack of regular in-person interaction may postpone emotionally charged conversations, and even when partners reunite, their limited time together can cause unresolved issues to be overlooked in favor of maintaining harmony [26]. This finding adds a new perspective to existing literature on gender and emotion regulation [58–60], highlighting the importance of relationship context in shaping how emotional suppression manifests across genders.

Another key finding of this study was the higher prevalence of somatic symptoms among individuals in LDRs compared to those in GCRs. This aligns with research suggesting that somatic symptoms can be influenced by a range of psychological factors, including childhood trauma, insecure attachment [53], negative emotions, rumination, avoidance, and health concerns [61]. Unique stressors in LDRs—such as physical separation, uncertainty about the future, and limited communication—may contribute to this elevated risk [26]. The absence of regular face-to-face interaction can increase vulnerability to daily stressors and intensify physiological responses. In addition, difficulties in maintaining emotional closeness and coping with prolonged absences may further undermine physical well-being [11].

These findings are consistent with Pereira et al. [14], who reported that 83% of women in LDRs experienced at least one moderate to severe somatic symptom, which negatively impacted life satisfaction. However, despite such evidence, few studies have directly compared the prevalence of somatic symptoms between LDRs and GCRs. Our results suggest that emotional ambivalence—found at higher levels in the LDR group—may partially explain the increased physical complaints observed in these relationships.

In conclusion, although long-distance relationships differ from geographically close ones in terms of responsibilities, changing roles [9], reduced physical and sexual contact [8], and diminished social support [62], the present study did not find a significant difference between the two groups in the mediation model of emotional suppression. The only notable group differences were related to gender patterns and the higher prevalence of somatic symptoms in LDRs. These findings suggest that being in a long-distance relationship does not inherently lead to emotional difficulties or impaired communication [63]. In fact, some research has shown that intimacy, love, and relationship satisfaction in LDRs can be comparable to—or even exceed—those in GCRs [64, 65]. For example, Merolla [66, 67] found that both LDR and GCR partners employ adaptive strategies to sustain their relationships, with LDR individuals often relying on these strategies more frequently due to distance-related challenges. Similarly, Du Bois et al. [50] reported no significant differences in relationship satisfaction and found that LDR participants sometimes engaged in healthier behaviors such as better diet and exercise.

These results highlight that overemphasizing objective aspects of distance—like physical separation—may obscure the importance of subjective experiences, such as perceived closeness, communication style, optimism about the relationship's future, and overall emotional satisfaction [63]. These subjective factors may be more influential in determining relational and psychological outcomes than physical proximity alone.

## Conclusion

This study aimed to examine the mediating role of emotional suppression in the relationship between ambivalence over emotional expression and somatic symptoms in individuals involved in long-distance and geographically close relationships. Although emotional suppression did not mediate this relationship, our findings still contribute meaningful insights into the field of emotion regulation and psychosomatic health. Consistent with prior literature, we confirmed the significant link between ambivalence over negative emotional expression and somatic symptoms, reinforcing the importance of emotional expression for physical well-being. Importantly, our results indicated that relationship type did not substantially alter this association.These findings provide preliminary evidence that may question the common assumption that long-distance relationships are inherently less emotionally fulfilling or more psychologically challenging. They also align with emerging literature suggesting that LDRs can maintain comparable levels of emotional and relational health to GCRs; however, these conclusions should be interpreted cautiously, given the study's cross-sectional design and convenience sampling.

We also observed gender-related patterns in suppression that varied by relationship type, suggesting that societal norms and relational dynamics may influence emotional strategies differently across contexts. In addition, somatic symptoms appeared more prevalent among individuals in LDRs, highlighting the need for tailored interventions to address somatic distress in these couples. Considering the sociocultural nuances of emotional regulation in Iran, this study provides a novel contribution by situating LDR research within a non-Western population.

In sum, our findings highlight the complex links between emotional ambivalence, emotional suppression, somatic symptoms, and relationship dynamics. They emphasize the importance of considering contextual and gendered aspects of emotion regulation. To our knowledge, this is the first study in Iran comparing LDRs and GCRs in this regard, adding a culturally grounded perspective to cross-cultural relationship research.

## Limitations and future directions

Our findings should be interpreted within the context of study limitations. First, the sample was gender-imbalanced, with a predominance of female participants. While this reflects trends in participant availability for psychological studies, it may limit the generalizability. Future studies should aim for more diverse representation. Second, certain structural aspects of LDRs (e.g., duration of separation, communication frequency) were not assessed. Investigating such relational variables in future studies would deepen our understanding of how contextual features shape psychological outcomes in LDRs. We also observed associations without clarifying underlying mechanisms, which future studies could explore through mediators such as stress, emotion regulation, or attachment styles. A further limitation of this study is its cross-sectional design, which prevents causal conclusions and tracking changes over time. Future longitudinal research is needed to clarify the direction of these associations. Finally, participants were classified into LDR and GCR groups based on the sample mean of the LDR Index scores, as the instrument does not provide an established cut-off point. While this approach allowed us to differentiate between groups, it is data-driven and sample-dependent, which may limit the generalizability of our findings.

## Supporting information

**S1 Data. Anonymized dataset of all study variables provided in CSV format.**
(CSV)

**S2 Data. Anonymized dataset of all study variables provided in SPSS (.sav) format.**
(SAV)

**S1 Text. README for S1 and S2 Data.** Describing all variables.
(DOCX)

## Acknowledgments

We sincerely thank all co-authors for their invaluable contributions to this study. Each author played an important role in shaping the final manuscript. We also extend our gratitude to the participants who gave their precious time to the research.

## Author contributions

**Conceptualization:** Nazanin Okati.

**Investigation:** Nazanin Okati, Leyla Rangamiztoosi, Maryam Gholipour.

**Methodology:** Nazanin Okati, Leyla Rangamiztoosi.

**Project administration:** Fariba Zarani.

**Supervision:** Fariba Zarani.

**Writing – original draft:** Nazanin Okati, Leyla Rangamiztoosi, Maryam Gholipour.

**Writing – review & editing:** Fariba Zarani.

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
