## [Decision Letter · Decision Letter 0]

5 Aug 2025

PMEN-D-25-00309

The relationship between ambivalence over the expression of emotions and somatic symptoms among Iranian long-distance and geographically close partners: the mediating role of emotional suppression

PLOS Mental Health

Dear Dr. Okati,

Thank you for submitting your manuscript to PLOS Mental Health. After careful consideration, we feel that it has merit but does not fully meet PLOS Mental Health’s publication criteria as it currently stands. Therefore, we invite you to submit a revised version of the manuscript that addresses the points raised during the review process.

We look forward to receiving your revised manuscript.

Kind regards,

Lambert Zixin Li, Ph.D.

Academic Editor

PLOS Mental Health

Journal Requirements:

https://journals.plos.org/mentalhealth/s/figures 

https://journals.plos.org/mentalhealth/s/figures#loc-file-requirements 

2. Please ensure that your Ethics Statement is available in its entirety at the beginning of your Methods section, under a subheading 'Ethics Statement'.

3. In the online submission form, you indicated that “All data supporting the findings of this study are available upon reasonable request from the corresponding author.”. 

3. Uploaded as supplementary information.

Additional Editor Comments (if provided):

Dear Authors,

Thank you for submitting your manuscript to PLOS Mental Health. After review, we invite you to submit a revised version addressing the comments provided by the reviewers.

While your study shows potential, the reviewers have identified several areas that require clarification or improvement. Please submit a point-by-point response detailing how each comment has been addressed, along with a revised version of the manuscript.

We look forward to receiving your revision.

Sincerely,

Lambert Zixin Li, PhD

PLOS Mental Health

Reviewers' comments:

Reviewer's Responses to Questions

**Comments to the Author**

1. Does this manuscript meet PLOS Mental Health’s publication criteria ? Is the manuscript technically sound, and do the data support the conclusions? The manuscript must describe methodologically and ethically rigorous research with conclusions that are appropriately drawn based on the data presented.

Reviewer #1: Partly

Reviewer #2: Yes

2. Has the statistical analysis been performed appropriately and rigorously?

Reviewer #1: Yes

Reviewer #2: I don't know

3. Have the authors made all data underlying the findings in their manuscript fully available (please refer to the Data Availability Statement at the start of the manuscript PDF file)?

Reviewer #1: Yes

Reviewer #2: Yes

4. Is the manuscript presented in an intelligible fashion and written in standard English?

Reviewer #1: Yes

Reviewer #2: Yes

5. Review Comments to the Author

Reviewer #1: Lines 100- 101: Please provide the unit for the ages

Line 106: Was there a reason why you limited your participants to 45 years? Please shed light on this.

Lines 116-119: Was there any benchmark you used in categorizing the participants to LDRs and GCRs? If there were, please state them for the readers.

Line 146: ‘Questionnaire’ should be ‘questionnaire’

Line 158: The numbering of your themes and sub-themes, such as ‘2.3.4’ in the manuscript, is not necessary. It seems that you are formatting a thesis for publication. Please delete the numbers for consistency in this journal.

Line 195: Please revise your table title to: Participants’ demographic and descriptive statistics

Line 195: In the table, revise the unit of age as ‘years’

Line 197: Please include the age unit

Line 198: Please change ‘Chi2’ to ‘X2’, or get the actual symbol

Line 201: Please insert the time unit

Lines 347- 350: ‘This challenges common assumptions….’, you would need to scale down the tone of your argument because of your study design (cross-sectional) and sampling technique (convenience).

Lines 338- 369: Your conclusion is too long. Kindly abridge it.

Line 370: Also, include your study design (cross-sectional) as a limitation.

Reviewer #2: Introduction

You mentioned "few studies have explored these relationships in LDRs versus GCRs, particularly in non-Western populations" (lines 83), but you could clarify why Iranian couples are especially important to study (e.g., cultural norms around emotion expression, relationship expectations, or stressors etc.) It will provide a robust reason behind conducting the study.

Participants and Procedure

The phrase “The data were accessed from June 1, 2023” (line 110) is unclear. Likely meant “data collection began June 1, 2023?” because in line 105 you have mentioned that you began to collect data. Please rephrase it or clarify or remove the sentence.

Please mention how many missing data were there before cleaning.

Acknowledgement

please acknowledge the coauthors and their contribution. Also acknowledge the research assistants who worked for the study but could not be added as coauthors in the study

thank you

good luck

6. PLOS authors have the option to publish the peer review history of their article (what does this mean? ). If published, this will include your full peer review and any attached files.

**Do you want your identity to be public for this peer review?** For information about this choice, including consent withdrawal, please see our Privacy Policy .

Reviewer #1: No

Reviewer #2: No

---

## [Editor Report · Decision Letter 1]

29 Sep 2025

The relationship between ambivalence over the expression of emotions and somatic symptoms among Iranian long-distance and geographically close partners: the mediating role of emotional suppression

PMEN-D-25-00309R1

Dear Ms Okati,

We are pleased to inform you that your manuscript 'The relationship between ambivalence over the expression of emotions and somatic symptoms among Iranian long-distance and geographically close partners: the mediating role of emotional suppression' has been provisionally accepted for publication in PLOS Mental Health.

Best regards,

Lambert Zixin Li, Ph.D.

Academic Editor

PLOS Mental Health

Dear Authors,

Thank you for addressing the reviewers' comments. I invited the same reviewers to re-review your revised manuscript, but unfortunately none of them responded in time. To avoid further delay, I decided not to invite additional reviewers and instead carefully reviewed your manuscript and response memo myself. I found your paper to be of publishable quality.

Best regards,

Lambert Zixin Li